# A Novel Bayesian Framework Infers Driver Activation States and Reveals Pathway-Oriented Molecular Subtypes in Head and Neck Cancer

**DOI:** 10.3390/cancers14194825

**Published:** 2022-10-03

**Authors:** Zhengping Liu, Chunhui Cai, Xiaojun Ma, Jinling Liu, Lujia Chen, Vivian Wai Yan Lui, Gregory F. Cooper, Xinghua Lu

**Affiliations:** 1Department of Biomedical Informatics, School of Medicine, University of Pittsburgh, Pittsburgh 15206, PA, USA; 2School of Medicine, Tsinghua University, Beijing 100190, China; 3Department of Engineering Management and Systems Engineering, Missouri University of Science and Technology, Rolla, MO 65409, USA; 4Department of Biological Sciences, Missouri University of Science and Technology, Rolla, MO 65409, USA; 5Georgia Cancer Center, and Department of Medicine, Medical College of Georgia, Augusta University, Augusta, GA 30912, USA; 6UPMC Hillman Cancer Center, University of Pittsburgh Medical Center, Pittsburgh, PA 15232, USA

**Keywords:** HNSCC, cancer drivers, causal inference, cellular signaling, subtyping, tumor-specific, HPV infection

## Abstract

**Simple Summary:**

Numerous factors, such as genomic mutations, chromosomal changes, transcriptional controls, phosphorylation, and protein–protein interactions, among others, can affect the activation status of proteins. Although each data type only partially reveals the status of a particular gene’s disruption, downstream expression changes ultimately indicate the functional effects of cancer driver protein alterations. By combining data on transcriptome and genomic alterations, we have developed a Bayesian framework to infer driver activation state, and further tested our method to highlight both statistical and biological significance by applying our model to TCGA HNSCC patient data.

**Abstract:**

Head and neck squamous cell cancer (HNSCC) is an aggressive cancer resulting from heterogeneous causes. To reveal the underlying drivers and signaling mechanisms of different HNSCC tumors, we developed a novel Bayesian framework to identify drivers of individual tumors and infer the states of driver proteins in cellular signaling system in HNSCC tumors. First, we systematically identify causal relationships between somatic genome alterations (SGAs) and differentially expressed genes (DEGs) for each TCGA HNSCC tumor using the tumor-specific causal inference (TCI) model. Then, we generalize the most statistically significant driver SGAs and their regulated DEGs in TCGA HNSCC cohort. Finally, we develop machine learning models that combine genomic and transcriptomic data to infer the protein functional activation states of driver SGAs in tumors, which enable us to represent a tumor in the space of cellular signaling systems. We discovered four mechanism-oriented subtypes of HNSCC, which show distinguished patterns of activation state of HNSCC driver proteins, and importantly, this subtyping is orthogonal to previously reported transcriptomic-based molecular subtyping of HNSCC. Further, our analysis revealed driver proteins that are likely involved in oncogenic processes induced by HPV infection, even though they are not perturbed by genomic alterations in HPV+ tumors.

## 1. Introduction

Cancer is a complex disease characterized by uncontrollable cell growth and metastasis. The development of tumor cells is mainly caused by perturbation of oncogenic signaling pathways due to various somatic genome alterations (SGAs), such as mutations, copy number alterations and epigenetic changes [1,2,3]. As of today, we know that genetic perturbations and their impact on cancer signaling pathways vary across cancer types and even among individual tumors of the same tissue type [4,5]. Identifying cancer driver genes and inferring the activation states of their protein products in individual tumors may enable the true understanding of the molecular pathogenic mechanism in individual tumors, which may facilitate the development of mechanism-specific therapies for individual patients [6,7,8,9,10].

Head and neck squamous carcinoma (HNSCC) is an aggressive and heterogeneous disease, ranked the sixth most common cancer worldwide [11]. Several risk factors contribute to oncogenesis of HNSCC, including alcohol consumption, smoking and human papillomavirus (HPV) infection. Over the past decade, the rapid emergence of multi-omics data has enabled genomic research on systematically identifying genetic markers and cancer driver genes for HNSCC patient molecular subtyping. Chung et al. first introduced four molecular subtypes (basal, mesenchymal, atypical, and classical) based on the gene expression pattern in 60 HNSCC tumors and were able to use this first subtyping method to predict lymph node metastasis. They also identified driver genes enriched in these subtypes, which may help future research in therapy discovery [12,13]. Similarly, the Cancer Genome Atlas (TCGA) group classified HNSCC patients into four groups based on expression data and investigated the association between these four groups and clinical factors and single gene alterations [14]. These two major studies revealed high degree of HNSCC heterogeneity in terms of molecular phenotypes, likely cellular/molecular mechanisms for tumorigenesis among HNSCC. Despite these past efforts, it remains a major challenge to integrate multi-omics data to tease out the exact disease mechanisms of individual tumors, including the understanding of which particular SGAs in a tumor are drivers for oncogenesis and which pathways are perturbed by these driver SGAs in a tumor. Equipped with such a mechanistic understanding for individual tumors, it would be possible to group HNSCC tumors bearing similar disease mechanisms and help guide therapy design for treating individual tumors.

In this work, we extended our work on tumor-specific causal inference (TCI) and developed a novel Bayesian framework to characterize the activation status of cancer driver genes in a context specific manner (Figure 1) [8]. We used the TCI model to identify major driver genes among TCGA HNSCC tumors and their target differentially expressed genes (DEGs) in HNSCC. Then, based on TCI-inferred causal relationship between driver genes and DEGs, we developed an expectation-maximization (EM) algorithm [15] to infer protein activation states for these driver proteins in individual tumors (Figure 1b). Through mining the patterns of driver protein activation states, we identified four HNSCC molecular subtypes that are informative of patient outcomes while orthogonal to previous subtyping. Our method provides a novel framework of understanding the mechanisms of tumorigenesis of HNSCC that cannot be directly detected using either SGA or transcriptomic data alone.

## 2. Materials and Methods

### 2.1. Data Collection and Preprocessing

In this work, we collected genomic data, i.e., mutation, copy number alteration and gene expression, for a cohort of 5097 tumor samples across 16 different cancer types from TCGA, which were all obtained from the Xena platform [16]. First, we combined mutation data and copy number alteration data (GISTIC2) as somatic genome alterations (SGAs), such that a gene in a given tumor was designated as altered if it was affected by either an SM event and/or an SCNA event. Then, we determined differentially expressed genes, i.e., DEGs, by comparing the gene expression in the tumor cell against that in the corresponding tissue-specific normal cells. We assumed that the expression of each gene (log 2 based) followed Gaussian distribution in normal cells for a specific tissue type, and calculated the *p* values of each gene in a tumor to estimate whether the gene expression was significantly different in that tumor from the normal distribution of the normal cells (*p* < 0.005). We thus identified the DEGs for each tumor and created a tumor-gene binary matrix where 1 represents expression change and 0 represents no expression change.

### 2.2. Bayesian Framework: 1. Tumor-Specific Causal Inference (TCI) Method

Tumor-specific causal inference (TCI) is a novel Bayesian-based framework developed by our group that infers causal relationships between genome alterations and molecular phenotypic changes by integrating heterogeneous genomic data types [8]. Let **T** denote the tumor set; let **SGA_t_** denote a subset of genes with genome alterations in tumor *t* (i.e., the SGAs); let **DEG_t_** denote a subset of genes that are differentially expressed in tumor *t* (i.e., the DEGs). We assumed that a molecular phenotype change (e.g., a DEG) observed in a specific tumor should be caused by one of the SGAs observed in the tumor or a non-specific cause denoted as A_0_. TCI searches for the tumor-specific causal model *M_t_* with a maximal posterior probability P(*M_t_*|*D*) given the dataset *D*, i.e., SGAs and DEGs. In a tumor *t*, TCI scores an arc *A_h_* → *E_i_* between SGA *A_h_* and DEG *E_i_* based on the posterior probability of the arc, using a Bayesian framework as follows:(1)$$P\left(A_{h}\to E_{i}|D\right)=\frac{1}{Z}P\left(A_{h}\to E_{i}\right)\;P\left(D|A_{h}\to E_{i}\right)$$
where
(2)$$Z\;=\sum_{j=0}^{n\left(SGA\right)}P\left(A_{j}\to E_{i}\right)\;P\left(D|A_{j}\to E_{i}\right)$$
is a normalization term. Equation (2) shows that a potential causal SGA *A_h_* only competes with other SGAs observed in the same tumor to explain a molecular phenotype *E_i_*.

As shown in Equation (1), TCI involves two parts: the prior probability that *A_h_* causes *E_i_*, i.e., P(*A_h_* → *E_i_*), which can be evaluated at a population-level prior to observing current tumor t, and the conditional probability (aka the marginal likelihood) of data *D*, P(*D*|*A_h_* → *E_i_*), which assesses the functional impact of the causal edges. An important innovation of TCI is the procedure for evaluating P(*D*|*A_h_* → *E_i_*), which consist of assessing how well *A_h_* explains the variance of *E_i_* in tumors hosting *A_h_*, as well as how well the variance of *E_i_* is explained in tumors do not host *A_h_*. Therefore, due to the distinct composition of SGA set for different tumors, the tumor-specific conditional probability P(*D*|*A_h_* → *E_i_*) for the same causal edge can be different in different tumors, which is shown as follows:(3)$$P\left(D|M\right)=\prod_{i=1,\;n}P(D_{hi}|A_{h}\to E_{i})=\prod_{i=1}^{n}\prod_{j=1}^{q_{i}}\frac{\Gamma \left(\alpha_{ij}\right)}{\Gamma \left(\alpha_{ij}+N_{ij}\right)}\prod_{k=1}^{r_{i}}\frac{\Gamma \left(\alpha_{ijk}+N_{ijk}\right)}{\Gamma \left(\alpha_{ijk}\right)}$$
where *j* indexes over the states of the cause of *E_i_* in *M* (i.e., some variable *A_h_*); *q_i_* is the number of possible values of *A_h_* (in our case, it is 2, because *A_h_* is modeled as a binary variable); *k* is variable which indexes over the states of *E_i_*; *r_i_* denotes the total possible states of *E_i_* (in our case, it is set to 2); *N_ijk_* is the number of tumors in dataset *D* in which node *E_i_* has value *k* and its cause *A_h_* in *M* has the value denoted by *j*; *α_ijk_* is a parameter in a Dirichlet distribution that represents prior belief about P(*E_i_*|*cause*(*E_i_*)); Γ is the gamma function; $N_{ij}=\sum_{k}^{r_{i}}N_{ijk}$; and $\alpha_{ij}=\sum_{k}^{r_{i}}\alpha_{ijk}$. Detailed information of TCI method can be found in our original TCI paper [8].

### 2.3. Discovery of TCI Drivers and Target DEGs in HNSCC Patients

We first extracted TCI inferred SGA → DEG relationship for all HNSCC patients. We designated an SGA event in a tumor as a driver if it had 5 or more causal edges to DEGs that are each assigned a TCI *p*-value ≤ 0.05. For an SGA *A_h_*, we calculated its driver call rate as the ratio of number of tumors in which *A_h_* is designated a driver by TCI over the number of tumors in which *A_h_* occurs. We identified a total of 64 such driver genes with a driver call rate of 50% or greater in HNSCC patients. For a DEG *D_k_*, we calculated its DEG call rate as the ratio of number of tumors in which *D_k_* is designated as target DEG of any of 64 drivers over the number of tumors in which *D_k_* occurs. We identified a total of 903 such genes with a DEG call rate of 50% or greater in HNSCC patients.

### 2.4. Bayesian Framework: 2. Inference of Tumor Specific Driver Activation State Using Expectation Maximization (EM) Algorithm

For the second part of our Bayesian framework, we inferred the tumor-specific driver activation state using the expectation maximization (EM) method, which is designed to predict the values of hidden variables based on observed variables [17]. In the context of our model, we defined **SGA_t_** and **DEG_t_**, which are two sets of discrete observation variables that describe the state of SGAs and DEGs in tumor *t*.

Let **PRO_t_** denote the hidden variable, namely, protein activation state of the TCI drivers that we set out to infer from **SGA_t_** and **DEG_t_** in tumor *t*. We assumed that the values of each gene/protein in **SGAs** and **PROs** followed a Bernoulli distribution, only taking the value of 0 or 1. DEGs that are regulated by a gene in **PRO_t_** are the same as DEGs that are the TCI inferred targets of the corresponding gene in **SGA_t_** and values of each gene in **DEG_t_** follows a Gaussian distribution. The total number of HNSCC samples is *N* and *t* indexes over the tumor samples included in the HNSCC tumor set.

Parameters Initialization: We first used **SGA_t_** values as initial values of **PRO_t_** and calculate prior probability of driver *j* being active in tumor *t* as follows,
(4)$$P\left(PRO_{j}=1\right)=\frac{\sum_{t=1}^{N}PRO_{jt}}{N}$$
where *E_jkt_* represents the expression value of a gene, *k*, which is regulated by driver *j*, in tumor *t*, where *k* indexes over the genes that are TCI inferred target DEGs of driver *j*.

The expression value *E_jkt_* conditioning on protein activation state of driver *j* in tumor *t* follows a Gaussian distribution as shown in equation Equations (5) and (6).
(5)$$\begin{array}{c} P\left(E_{jkt}\;|\;PRO_{jt}=1\right)=\frac{1}{\sqrt{2\pi }\sigma_{jk}^{1}}\exp\left(-\frac{{\left(E_{jkt}-\mu_{jk}^{1}\right)}^{2}}{2{\left(\sigma_{jk}^{1}\right)}^{2}}\right) \end{array}$$
(6)$$\begin{array}{c} P\left(E_{jkt}\;|\;PRO_{jt}=0\right)=\frac{1}{\sqrt{2\pi }\sigma_{jk}^{0}}\exp\left(-\frac{{\left(E_{jkt}-\mu_{jk}^{0}\right)}^{2}}{2{\left(\sigma_{jk}^{0}\right)}^{2}}\right) \end{array}$$
where expression of gene *k* follows a Gaussian distribution with mean as $\mu_{jk}^{1}$ and standard deviation as $\sigma_{jk}^{1}$ when protein activation state of driver *j* equals 1, and follows a Gaussian distribution with mean as $\mu_{jk}^{0}$ and standard deviation as $\sigma_{jk}^{0}$ when protein activation state of driver *j* equals 0.

Expectation Step (E step): We calculated the probability that driver *j* is active based on the expression profile of its target DEGs in tumor *t* as follows,
(7)$$P\left(PRO_{jt}=1|DEG_{jt}\right)=\frac{P\left(PRO_{j}=1\right)\prod_{k=1}^{n\left(DEG_{jt}\right)}P(E_{jkt}|PRO_{j}=1)}{\sum_{i\in \left\{0,1\right\}}P\left(PRO_{j}=i\right)\prod_{k=1}^{n\left(DEG_{jt}\right)}P(E_{jkt}|PRO_{j}=i)}$$
where *DEG_jt_* denotes the expression profile of target DEG set of driver *j* in tumor *t*. Then, $P\left(PRO_{jt}=1|DEG_{jt}\right)$ was discretized if its value is larger or equal to 0.5.

Maximization Step (M step): After initial calculation and determination of $P\left(PRO_{jt}=1|DEG_{jt}\right)$, we updated the parameters in Equations (4)–(6) and started EM iteration. Then, we used $P\left(PRO_{jt}=1|DEG_{jt}\right)$ to test convergence. If the absolute difference between the marginal probability between two iterations was less than 10^−4^, we terminated EM and calculated the final prediction of protein activation states.

### 2.5. Patient Subtyping Using Consensus Clustering

We performed hierarchical clustering method to calculate distance between each pair of patients based on their DEG or SGA values, and divided patients into several groups based on their distance. We used the seaborne python package (https://seaborn.pydata.org/, accessed on 10 April 2022) and selected Euclidean distance as parameters for hierarchical clustering analysis. Since the hierarchical clustering method is deterministic, we performed 100 subsampling with a factor of 0.8 for all patient cases. We further used consensus clustering to decide number of patient groups based on 100 hierarchical clustering results. Consensus clustering is a robust clustering method that utilize multiple iterations of hierarchical clustering method. For each cluster number *K*, we created a consensus matrix calculate the cumulative distribution function (CDF). Then, we decided the number of clusters as 4 based on the area under CDF. We used ConsensusClusterPlus R package for consensus clustering analysis (https://bioconductor.org/packages/-release/bioc/html/ConsensusClusterPlus.html, accessed on 15 April 2022).

### 2.6. Reconstruction of Driver → DEG Causal Network

To assess whether the SGA → DEG causal relationship is activating or repressing, we calculated the point-biserial correlation between each driver and its target DEGs. We constructed an *r* × *c* SGA → DEG regulatory matrix M, where *r* is the number of drivers and *c* is the number of target DEGs and *m_ij_* ∈ {0, 1, −1} denotes a causal relationship between SGA *i* and DEG *j* can be none, activating or repressing. We then performed ISA2 method to identify biclusters of SGAs and their co-activated/repressed DEGs [18,19]. We used consensus clustering to identify the most significant driver clusters, DEG clusters, and driver-DEG clusters from 100 ISA2 iterations because ISA2 is a probabilistic algorithm that produces somewhat different results on different runs due to random initialization (Appendix A).

## 3. Results

### 3.1. TCI Method Can Identify Major Cancer Driver Genes and Their Causative DEGs Targets for HNSCC Patients

We previously developed a Bayesian causal discovery framework, i.e., TCI [8], to characterize the functional impact of SGAs (combination of gene mutations and copy number alterations) in regulating DEGs of individual tumors by integrating multiple genomic data types. We collected genomic data of 5097 tumors across 16 cancer types from TCGA, and we applied TCI to infer causal relationship between SGAs and DEGs for each tumor. Then, we collectively identified over 600 significant cancer drivers that regulate DEGs in tumors of one or more cancer types (Figure 1a). To minimize false discovery rate, we required that driver call rate and DEG call rate to be greater than or equal to 0.5. A driver call rate is defined as the ratio of number of tumors in which an SGA is designated a driver by TCI over the number of tumors in which the SGA occurs. A DEG call rate is defined as the ratio of number of tumors in which a DEG is designated as target DEG of any of drivers over the number of tumors in which the DEG occurs. By limiting TCI inferred SGA → DEG relationship discovered in HNSCC tumors, we identified 64 driver genes with a driver call rate of 50% or greater and 903 DEGs designated as targets of any of 64 drivers with a DEG call rate of 50% or greater in HNSCC tumors (Methods).

Among these 64 HNSCC drivers identified by our TCI, majority are well-known drivers including *EGFR*, *TP53*, *PIK3CA*, *NOTCH1*, *FAT1*, *HRAS*, *CDKN2A* and *NFE2L2* [14,20,21]. Our method also revealed novel HNSCC drivers, which have been experimentally validated in recent studies. For example, *ZNF703E* copy number variation was found to be associated with its overexpression in HNSCC tumors shown by quantitative real time PCR, and over-expression of *ZNF703E* indicated poorer patient survival vs. non-overexpression ones [22]. *LRP1B* mutation was reported to be associated with HPV status and poor disease outcome [23]. *MIR548K* was identified as one of the seven miRNAs that are associated with poor prognoses of HNSCC [24]. Pharmacologically inhibiting *KEAP1* in HNSCC cells was found to enhance radiosensitivity of HNSCC cells [25]. *ARID1A* was identified to be a tumor suppressor and a tumor stemness repressor in HNSCC cells [26].

The majority of the 64 TCI-identified drivers are cancer drivers or candidates related with carcinogenesis processes, and we determined their target DEGs to illuminate the underlying HNSCC pathways or biological processes. Using Reactome pathway analysis on these 903 TCI-derived target DEGs that are significantly associated with 64 TCI-derived drivers, we identified 59 significantly enriched pathways including 104 DEGs (Figure 2a,b, Appendix A). For example, Collagen production plays an important role in the development of fibrosis and is a prominent component of the tumor microenvironment [27]. Our DEGs are shown to be enriched in Collagen biosynthesis pathways—Collagen formation (*p*.adjust = 3.4 × 10^−12^), Collagen biosynthesis and modifying enzymes (*p*.adjust = 8.5 × 10^−9^), Assembly of collagen fibrils and other multimeric structures (*p*.adjust = 5.6 × 10^−7^), etc. DNA damage created by environmental factors such as tobacco usage and alcohol consumption are found to be correlated with the increased HNSCC risk by multiple studies, but the molecular mechanism of DNA replication and repair pathway in resistance to therapy and treatment of HNSCC patients remains unclear [28,29,30]. Our DEGs are also found to be enriched in pathways related to DNA replication and repair processes—DNA strand elongation pathway (*p*.adjust =1.6x10^−11^), Extension of Telomeres (*p*.adjust = 4.4 × 10^−7^), lagging strand synthesis pathway (*p*.adjust = 4.4 × 10^−7^), etc. As shown in Figure 2b, TCI genes involved in these two major groups of enriched pathways have distinct expression patterns and are regulated by a diverse collection of TCI inferred drivers except for *TP53* and *LRP1B*. *TP53*, the most commonly mutated gene in HNSCC patients, induces cell cycle arrest and apoptosis, and also up-regulates collagen gene expression to inhibit angiogenesis [24,28,29]. *LRP1B*, a member of the LDL receptor family of lipoprotein receptors, has recently been proposed as a putative tumor suppressor, with functions related to cell cycle arrest and modulation of cell migration/spreading [31,32,33].

We then set out to reconstruct the driver → DEG regulatory network of all 64 TCI drivers and 903 DEGs for HNSCC patients, in addition to pathway gene analysis. To begin, we calculated the point-biserial correlation between each driver and its target DEGs to determine activating/repressing causal interactions. Calculating the point-biserial correlation is equivalent to calculating the Pearson correlation between a continuous and a dichotomous variable. Then, we applied ISA2 bi-clustering method and consensus clustering method to divide drivers and DEGs into various groups (Method, Appendix A). Figure 2c illustrates the causal network structure between drivers and DEGs, with DEGs separated into 11 groups and each group is driven by a distinct set of drivers. Genes in the top right corner, for example, are heavily involved in cell cycle G1/S phase. TCI drivers, i.e., *RB1*, *ARID1A* and *FBXW7*, positively control their expression, which is experimentally supported by literature evidence [33,34,35].

### 3.2. TCI-Derived Molecular Profiles Predict Significant Prognostic Outcome Differences among HNSCC Patient Subtypes

HNSCC is a genomic disease of high degree of inter-tumor heterogeneity, in terms of genomic alterations and molecular/cellular phenotyping. It is an attempting goal to discover patterns of genomic alterations and discover subtypes of HNSCC with common disease mechanisms. As a test, we directly used all SGAs data as input features and grouped TCGA HNSCC tumors into four clusters—where the number of clusters was pre-defined per previous studies [10,11,12]—according to similarity of SGA profiles of the tumors (Figure 3a). However, no clear patterns can be visually discerned according to this analysis. Further examining the survivals of these patient groups did not show significant differences among the groups (*p* = 0.71).

We then set out to examine whether the 64 drivers identified by the TCI are more informative of disease mechanisms and patient outcomes. Using genomic status of the TCI-derived driver genes as input features, we were able to reveal certain patterns of genome alterations of tumors which segregate patients into groups with more significant survival outcome differences (*p* = 1.5 × 10^−2^) (Figure 3b). Visually inspecting the figure, one can see certain SGAs events co-occur in tumors within a common cluster. Thus, the results indicate that SGA status of 64 drivers are informative for HNSCC patient subtyping.

Since SGA affects oncogenesis through perturbing signaling pathways, and a common signaling pathway can be perturbed by distinct SGAs in different tumors, we hypothesized that transcriptome regulated by SGA-perturbed pathways can be more informative of disease mechanisms of tumors. We then performed clustering analyses of the TCGA HNSCC tumors, using all 20k DEGs observed in HNSCC tumors and 904 TCI-derived driver-targeted DEGs as input features, and we divided tumors into four subtypes (Figure 3c,d). It is interesting to note that more clearer patterns were revealed by the driver-targeted DEGs; moreover, patient groups derived using these features exhibited much more significant differences in survival outcomes (*p* = 1.4 × 10^−3^). This result suggests that the 903 DEGs causally regulated by the 64 TCI drivers provided a concise and effective representation of HNSCC tumors’ characteristics in terms of pathway perturbation by the driver SGAs.

We compared our four HNSCC subtypes with previously reported TCGA HNSCC molecular subtyping and check the patient overlaps between two studies. Survival analysis shows that both studies have categorized patients into groups with significant prognostic outcomes, but our subtyping is orthogonal to TCGA HNSCC molecular subtyping (Appendix A). While 32 out of 46 tumors in our TCI-derived Subtype 1 are categorized as Atypical in TCGA paper, Atypical tumors are also distributed in our subtype 1, 2, and 4, indicating our methods has divide the Atypical group. Patients in the TCI-derived subtype 2 mainly correspond to the Classical subtype defined in TCGA study, i.e., 37 out of 76, and Atypical, i.e., 22 out of 76. Patients in subtypes 3 and 4 mainly contain patients from TCGA Basal, i.e., 40 out of 116 for subtype 3 and 27 out of 47 for subtype 4, and Mesenchymal, i.e., 45 out of 116 for subtype 3 and 16 out of 47 for subtype 4. The patient overlap ratio between our subtype 3 and 4 and TCGA Basal and Mesenchymal is 70%. The survival outcomes are not significantly different between patients in the TCGA Basal and Mesenchymal classes (Appendix A). However, based on our TCI-derived subtyping, we are able to divide the same group of patients into two groups with significant prognostic outcomes as shown in Appendix A.

We then examined the association between the 903 DEG-derived four patient subgroups and known HNSCC risk factors to examine whether our subtyping reflect disease mechanism of HNSCC. As shown in Figure 3e, the number of male patients is about three times as female patients in our cohort. Group 3 and 4 with more female patients, i.e., 38% and 31%, have worse survival outcomes than group 1 and 2, i.e., 19% and 17% (*p* = 0.21). Majority of patients reported a history of smoking, but there is no correlation between smoking frequency and subsequent survival outcome in four patient subgroups. Alcohol consumption level is also similar among four patient groups (*p* = 0.78). For molecular features, we notice that patient group 1 are significantly enriched with HPV negative patients, i.e., ~70%, and the patients in this group had the best survival outcome. We also compared mutation status of three genes, i.e., *TP53*, *NOTCH1*, *NSD1*, in the four patient groups. *TP53* is the most frequent mutated gene in HNSCC, but *TP53* mutation rate is much lower in group 1 (27%) than in other groups (group 2: 92%, group 3: 76%, group 4: 82%). *NOTCH1* is the second most frequent mutated genes in HNSCC, and its mutation rate is lower in patient group 1 (12%) and 2 (18%) than in group 3 (27%) and 4 (20%) as shown in Figure 3e (*p* < 0.05). The *NSD1* gene is a tumor suppressor genes and is associated with HNSCC patient survival outcome [36]. *NSD1* mutation is mainly enriched in patient group 2 (33%) compared to the other three groups (group 1: 10%, group 3: 6%, group 4: 4%), as shown in Figure 3b (*p* < 0.01), indicating that it may play an important role in cancer development of this subtype of tumors. As shown in Figure 3f, there is no significant difference in age and tumor mutational burden (TMB) levels across patient groups.

### 3.3. Infer Driver Activation States and Represent HNSCC Tumors in the Space of Cellular Signaling Systems

Because different SGAs can perturb a pathway in different tumors, we reasoned that when a pathway is disrupted by SGA in one member or activated by cross talks from other pathways, member proteins’ functional states likely are active. In other words, even if a driver protein is not perturbed by an SGA event, it can be in an active state due to signals from upstream or crosstalk with other pathways. The active states of pathways (and their member proteins) dictate a tumor’s disease mechanisms, hence gaining insight of the functional states of pathways and signaling proteins would be informative for precision oncology.

The TCI model provides a framework to infer the activation states of driver proteins by combining the genomic and transcriptomic data. Based on the causal relationships between SGAs and their target DEGs, we treat the functional states of driver proteins as latent variables, and we trained a statistical model using the expectation-maximization (EM) algorithm to infer the functional states of all 64 HNSCC-related driver protein in an individual tumor conditioning on the SGA and DEG data observed from the tumor (Method). As shown in Figure 4a, the introduction of EM algorithm enables us to represent each individual HNSCC tumor in the space of cellular signaling systems, by projecting transcriptome data into protein activity state. Based on the 64 drivers’ protein activation pattern in HNSCC tumors, we are also able to divide patients into four groups (named as the Protein Activation subgroups). The four subgroups categorized by using predicted protein activation pattern also shows significant survival outcome difference, i.e., *p* = 2.5 × 10^−3^ (Figure 4e), and the patient membership is in accordance with four subtypes categorized by 903 DEGs. This result suggests the integrity of molecular information is maintained during transformation from TCI identified 903 HNSCC gene expression to 64 TCI driver activities.

As shown in Figure 4b, 70.7% of patients in Protein Activation subtype 2 are HPV-positive associated with best survival outcomes, while only 1.5% of patients in Protein Activation subtype 3 are HPV-positive associated with much worse survival outcomes. HPV-positive patients have active drivers that are fundamentally different from HPV-negative patients, implying that HPV virus might have activated these driver proteins to cause HPV-associated pathway activities. Figure 4c illustrate that several drivers are predicted to be functionally more active in HPV-positive patients than in HPV-negative patients, while their SGA states are similar. For example, HPV infection activates the PI3K signaling pathway by modifying various molecular events to promote carcinogenesis, hence PI3K pathway plays a more critical role in HPV-positive HNSCC cancer [17]. However, the functional importance of *PIK3CA* in HPV-positive patients will obscured if based on its mutation rate, which is slightly greater in HPV-positive than in HPV-negative patients, i.e., 50% and 32%, respectively. Because our method inferred *PIK3CA* activity in HPV-positive patients without *PIK3CA* mutations, *PIK3CA* is inferred to be active, either mutated or functionally activated (presumably through pathway cross-talks), in 98.5% of HPV-positive HNSCC patients, comparing to 57.8% of HPV-negative HNSCC patients. As shown in Figure 4d, patients with *PIK3CA* mutations and inferred *PIK3CA* activity exhibit similar expression profile of *PIK3CA* target DEGs, whereas patients without *PIK3CA* mutations or inferred *PIK3CA* activity have completely distinct expression profile. Target DEGs of other drivers in Figure 4c have similar expression patterns for patients with and without driver mutations, as well as predicted driver activity (Appendix A).

## 4. Discussion

In this study, we have developed and evaluated a novel and effective approach for identifying signaling mechanisms by combining transcriptome and protein perturbations in individual HNSCC tumors. Our method can depict tumors in the space of cellular signaling system, whereas mutation data only show the perturbation status of a single gene. The TCI-derived drivers and their causal DEGs are found to be enriched in a variety of carcinogenesis pathways, and we are able to classify HNSCC patient tumors into four subgroups with significant survival differences. The novelty and potential utility of our methods are as follows.

First, the TCI method differentiates driver and passenger SGAs and further infers the causal relationship between SGAs and DEGs at the level of individual tumors. This provides researchers a way to study disease mechanisms of individual tumors, which in turn enables discovery of the tumor subtypes that share common disease mechanisms. On the other hand, the conventional molecular subtyping often utilizes the whole transcriptome, which can be significantly influenced by cell origins and other non-oncogenic factors influencing gene expression. Second, our method enables us to infer the “activation states” of all driver proteins in a tumor, even when such genes are not perturbed. Such information can be critical in the process of clinical decision making in precision medicine, laying the groundwork for drug sensitivity prediction and repurposing. Conventionally, application of a molecularly targeted drug is mainly guided by the mutation status of the targeted proteins. It can be conjecture that if a targeted protein is aberrantly activated in a tumor, the cancer cells may respond to a drug that targets the protein. This hypothesis remains to be tested in the future. Third, inferring the state of driver proteins will enable researchers to further tease out the signaling pathways by inferring the causal relationships among driver SGAs. For example, our results suggest that many driver proteins might have been aberrantly activated in the HPV+ tumors, even though their genes were not perturbed by SGA events. This indicates that HPV infection may activate these oncogenic proteins through previous unknown cross talks, such as protein–protein interactions. In summary, we anticipate that the capability of identifying driver genes and infer their protein activation states in tumors will have a broad impact on studying cancer disease mechanisms and guiding precision medicine. It would be interesting to further reconstruct cancer pathways based on driver status and target DEG expression profile.

## 5. Conclusions

By integrating transcriptome and genomic alterations data in individual HNSCC tumors, we successfully developed and assessed a unique and efficient Bayesian method for inferring the driver activation state. While mutation data only expose the perturbation status of a single gene, our technique can infer driver activation state and depict tumors in the context of cellular signaling systems. The TCI-derived drivers and their causal DEGs are found to be enriched in a variety of carcinogenesis pathways that are patient-specific, and we can classify HNSCC patient tumors into four subgroups with notable survival differences and biological significance.

Our comprehension of the mechanisms behind HNSCC disease and the advancement of tailored medicine will be significantly enhanced by the reconstruction of cancer pathways and the determination of their activity for each patient. The patient-specific active state of driver genes, some of which are therapeutic targets, could be used to predict drug sensitivity and provide insights into personalized therapy.

## Figures and Tables

**Figure 1 cancers-14-04825-f001:**
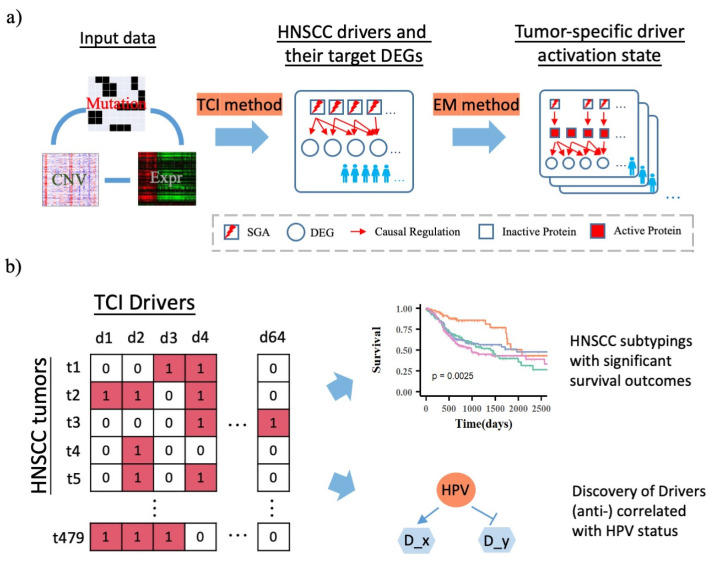
Workflow of the driver activation state inference, pathway analysis, subtyping and survival analysis, and driver-HPV status association analysis for HNSCC patients. (**a**). Integration of multi-omics data types to infer driver activation state inference using TCI method and EM method. Mutations and CNV data are combined into SGA data and expression data are transformed to DEG data, which are used as the input of the TCI model. TCI calculates casual relationships between SGAs and DEGs for each individual patient, and we further generalize the most statistically significant causal SGA → DEG interactions in TCGA HNSCC patient cohort. Based on TCI inferred SGA → DEG causal relationship and selected DEG expression profile, EM algorithm is used to infer protein activation states for each HNSCC patient. (**b**). HNSCC patients can be divided into four subtypes based on 64 driver activation state and their survival outcome are significantly different among these patient subtypes. Our analysis also shows the viral driver proteins can be triggered by HPV status and regulate its downstream gene expression.

**Figure 2 cancers-14-04825-f002:**
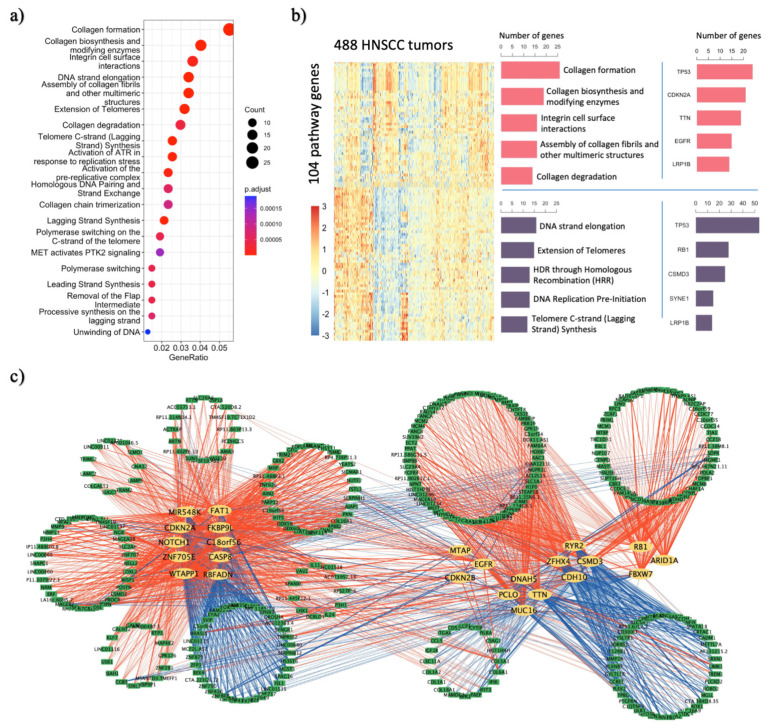
TCI result analysis for HNSCC patients. (**a**) Reactome pathway enrichment analysis of 903 TCI DEGs. Top 20 pathways with highest adjusted *p* values are listed. (**b**) 104 gene from 59 enriched Reactome pathways can be separated into 2 groups based on their expression profile. Forty-seven genes in the upper part are more associated with collagen biosynthesis processes with *TP53*, *CDKN2A*, *TTN*, *EGFR* and *LRB1P* as their major drivers. Fifty-seven genes in the lower part are more associated with cell cycle and DNA repair processes with *TP53*, *RB1*, *CSMD3*, *SYNE1* and *LRB1P* as their major drivers; (**c**) Driver and DEG causal relationship network. Each yellow node represents a TCI driver and each green node represents a DEG. Each red line represent a driver → DEG activating causal interaction and each blue line represents a repressing driver → DEG causal interaction.

**Figure 3 cancers-14-04825-f003:**
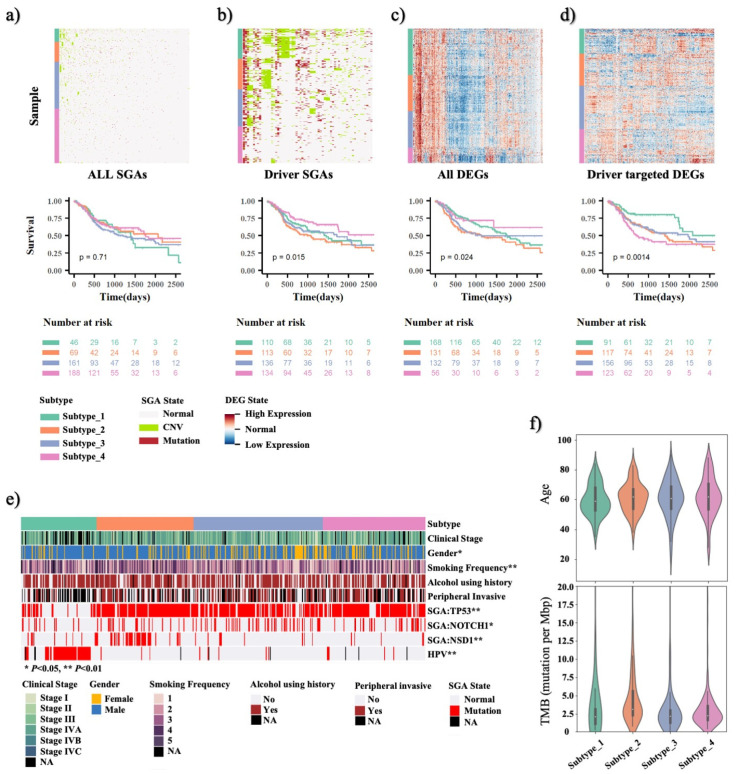
Molecular subtyping of HNSCC tumors using TCI-derived features. The subtypes of HNSCC patients are classified using hierarchical clustering and then studied using survival analysis using (**a**) all 17,646 SGA states, (**b**) 64 TCI SGA states, (**c**). all 10,120 gene expression levels and (**d**) 903 TCI-inferred DEG expression levelss, respectively. The colors of the side bar of the heatmap represent the subtype. Survival times are constrained to 2500 days. Survival analysis is carried out to calculated *p* value for patient groups categorized using different datasets. (**e**). Statistical association of clinical features with 4 HNSCC patient subtypes categorized using 903 TCI DEGs. NA in the legend represents missing values. (**f**). Violin plot of age and tumor mutation burden (TMB) distributions in 4 HNSCC patient subtypes categorized using 903 TCI DEGs.

**Figure 4 cancers-14-04825-f004:**
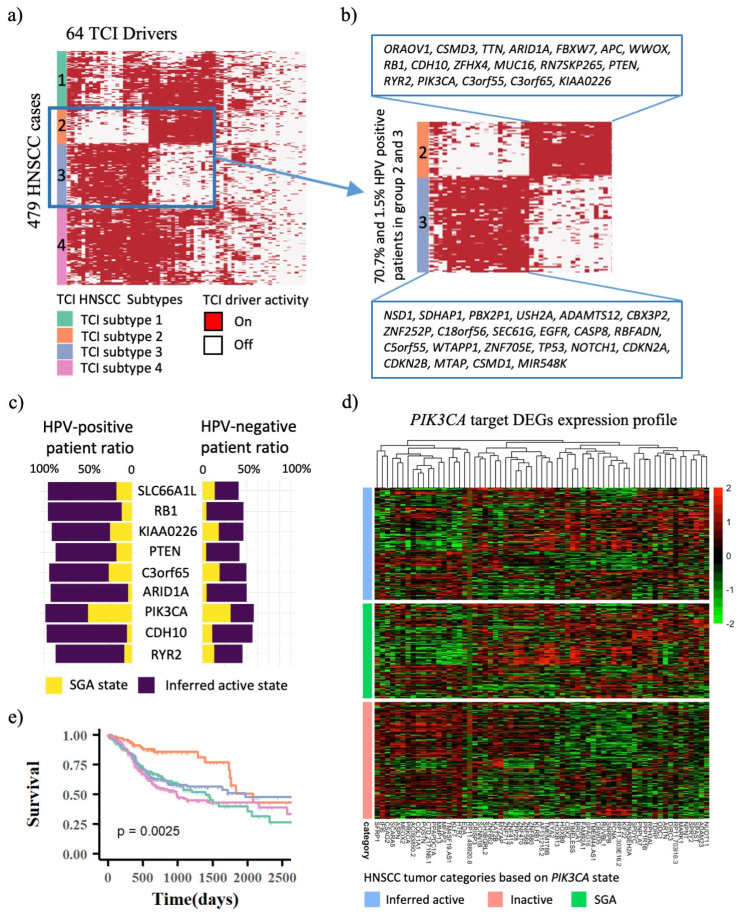
Clustering based on inferred protein states identified four subtypes and drivers found to be associated with HPV-positive patients. (**a**) Four subtypes of HNSCC samples were discovered based on EM inferred driver protein activation states, where red denotes inferred driver state is on and white denotes inferred driver state is off. (**b**) 70.7% of patients in subtype 2 are HPV-positive patients, while 1.5% of patients in subtype 3 are HPV-positive. Patients in subtype 2 and 3 are associated with different set of drivers. (**c**) Bar plot of drivers with higher inferred protein activation rate but similar SGA frequencies in HPV-positive than in HPV-negative patients, where yellow bar denotes the SGA frequency and purple bar denote the inferred driver activation rate. (**d**) Expression profiles of TCI-derived *PIK3CA* targeted DEGs in three groups of patients, i.e., patients with *PIK3CA* SGA, patients with inferred *PIK3CA* inferred protein activity and patients without *PIK3CA* SGA or inferred protein activity. (**e**) Survival analysis of four HNSCC patients subtypes categorized using inferred driver activation states.

## Data Availability

Molecular datasets including gene expression data, mutation data, copy number alteration data and methylation data of 488 TCGA HNSCC patients are collected from Xena database (https://xenabrowser.net/datapages/, accessed on 1 April 2020). Phenotype data including patients age, gender, drinking and smoking history, HPV status and survival data are also downloaded from Xena database. Python Codes for driver activation state inference are well documented and available at: https://github.com/Arcade0/HNSC-TCI.git, accessed on 25 September 2022.

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
