# Peer review of "A Novel Bayesian Framework Infers Driver Activation States and Reveals Pathway-Oriented Molecular Subtypes in Head and Neck Cancer"

_cancers, 2022, doi:10.3390/cancers14194825_

Round 1

Reviewer 1 Report

In this manuscript, Liu et al, described a multi-step framework, which a big part of this framework has been previously developed in a publication by the same authors, to infer driver activation state by combining genomic alteration and transcriptome data. Authors further applied this framework to a public dataset of head and neck  squamous cell carcinoma from TCGA. This is an interesting study and very well suited to be shared with the community. However, I have some minor questions/comments that I would like to be addressed prior to acceptance of the manuscript.

1- Please double check the language. There are couple of typos in the manuscript.

2- It seems result section is a mixture of both result and discussion. A lot of discussion point in the result section. Please separate the discussion from the result section.

3- If possible, please include the code in the final version. This way, the community can better implement the developed pipeline.

4- How did the authors calculate p value for 4 different survival groups in Figure 3? If this is a log rank test, then it is only applicable to two-group comparison. Please explain about the p value.

5- Page 11 (of 17), line 331. Why did authors choose 4 subtypes to divide the tumors into? What is the underlying reason for 4? if you change to to 5, are you missing the distinct behavior in the data?

Author Response

In this manuscript, Liu et al, described a multi-step framework, which a big part of this framework has been previously developed in a publication by the same authors, to infer driver activation state by combining genomic alteration and transcriptome data. Authors further applied this framework to a public dataset of head and neck  squamous cell carcinoma from TCGA. This is an interesting study and very well suited to be shared with the community. However, I have some minor questions/comments that I would like to be addressed prior to acceptance of the manuscript.

1- Please double check the language. There are couple of typos in the manuscript.
Answer: We thoroughly reviewed the manuscript and corrected the typos accordingly.

2- It seems result section is a mixture of both result and discussion. A lot of discussion point in the result section. Please separate the discussion from the result section.
Answer: Thank you for your comment. We moved discussion points from the results section to the discussion section if they did not aid in context comprehension (line 307 page 9).

3- If possible, please include the code in the final version. This way, the community can better implement the developed pipeline.
Answer: We uploaded our code to GitHub and made it publicly available to community. Link is provided in the “Code Availability” section of the paper.

4- How did the authors calculate p value for 4 different survival groups in Figure 3? If this is a log rank test, then it is only applicable to two-group comparison. Please explain about the p value.
Answer: It is possible to compare two or more survival curves using the log-rank test. The null hypothesis states that there is no difference in survival between the two groups. The log rank test is a non-parametric test that makes no survival distribution assumptions. It compares the observed number of events in each group to the number that would be expected if the null hypothesis were true. The log rank statistic can be approximately distributed as a chi-square test statistic for two or more groups. The function survdiff() in the "survival" R package computes log rank test by comparing two or more survival curves.

References:
https://cran.r-project.org/web/packages/survival/index.html

5- Page 11 (of 17), line 331. Why did authors choose 4 subtypes to divide the tumors into? What is the underlying reason for 4? if you change to 5, are you missing the distinct behavior in the data?
Answer: We performed consensus clustering analyses, which indicate that 4 clusters provide better separation of the clusters.  Also, based on the DEG profile and inferred driver activation state as shown in Figure 4a, visual inspection also supports four groups. The number of clusters was also pre-defined per previous TCGA HNSCC studies.

Reviewer 2 Report

The manuscript introduces a Bayesian framework that provides a systematic approach to 1) infer causal relationships between somatic genome alterations and expressed genes, and 2) a data fusion mechanism for various data sources  (genomic + transcriptomic) to infer functional activation of proteins, with insights on the HPC induced oncogenic processes.

Observations/suggestions:

- with a focus on methodology, did you also look at other methods in the realm of Bayesian approaches which combines structural learning of graphs models and inference of causal effects? For instance the work in https://doi.org/10.1007/s10260-021-00579-1 and how this work relates to your approach, especially when considering the TCI component. This can be further extended towards methods that combine structural learning and causal effect estimation, leading to a posterior distribution over the space of DAGs, DAG parameters and causal effects.

- when discussing about the network reconstruction in subsection 2.6 I would also raise the relevant point of considering if the magnitudes of coefficients in the model are predictive of genetic interaction profile similarities. There is some relevant work in: https://doi.org/10.15252/msb.20199174. This is motivated by the fact that we typically require datasets that elicit diverse physiological regulatory responses, and possess sufficient information to disambiguate the drivers of each regulatory response.

- there is strength in the framework's ability to infer the causal relationship between SGAs and DEGs at the level of individual tumours. Would be nice to understand how these insights could be combined with mechanistic modelling to improve therapy design?

- did you consider a dynamical systems-like analysis of the inferred “activation states”  of all driver proteins in a tumour extracted by your framework? This would be a very useful investigation in drug sensitivity prediction, and even better, in the tumour-immune-drug interactions modelling (see for instance https://doi.org/10.1101/2020.10.08.331678 and https://www.preprints.org/manuscript/202209.0012/v1)

Author Response

The manuscript introduces a Bayesian framework that provides a systematic approach to 1) infer causal relationships between somatic genome alterations and expressed genes, and 2) a data fusion mechanism for various data sources  (genomic + transcriptomic) to infer functional activation of proteins, with insights on the HPC induced oncogenic processes.

Observations/suggestions:

- with a focus on methodology, did you also look at other methods in the realm of Bayesian approaches which combines structural learning of graphs models and inference of causal effects? For instance the work in https://doi.org/10.1007/s10260-021-00579-1 and how this work relates to your approach, especially when considering the TCI component. This can be further extended towards methods that combine structural learning and causal effect estimation, leading to a posterior distribution over the space of DAGs, DAG parameters and causal effects.

Answer: The work reported in this study focuses on inferring the activation state of the driver from TCI DEG status. We are further investigating causal structure learning in a separate study. 

In terms of relationship with the TCI algorithm, our current study relies on the causal relationships inferred by TCI, thus can be thought as an extension of TCI.  TCI does not represent nor attempt to estimate the functional state of driver genes.  In contrast, this study concentrates on infer the functional state of driver proteins in the cases when their genes were not perturbed by SGA events.

- when discussing about the network reconstruction in subsection 2.6 I would also raise the relevant point of considering if the magnitudes of coefficients in the model are predictive of genetic interaction profile similarities. There is some relevant work in: https://doi.org/10.15252/msb.20199174. This is motivated by the fact that we typically require datasets that elicit diverse physiological regulatory responses, and possess sufficient information to disambiguate the drivers of each regulatory response.

Answer: This is a very good point. Unfortunately, no such data set exists for real cancer samples (like the HNSCC patients). We may evaluate our model performance and network reconstruction in a separate study using other datasets, such as the datasets cited in this MSB paper.

- there is strength in the framework's ability to infer the causal relationship between SGAs and DEGs at the level of individual tumours. Would be nice to understand how these insights could be combined with mechanistic modelling to improve therapy design?

Answer: Yes. In the future study, we plan to utilize tumor-specific inferred driver activation states to predict drug sensitivity. We hypothesized that when a pathway is disturbed by drivers in one member or activated by cross-talks from other pathways, the functional states of the member proteins are active. A driver can be in an active state even if it is not affected by its own SGA event due to signals from upstream or crosstalk with other pathways. The active states of pathways determine a tumor’s disease mechanisms; hence, having knowledge of the functional states of pathways and their signaling proteins/drivers would be important to predict drug sensitivity and improve therapy design.

- did you consider a dynamical systems-like analysis of the inferred “activation states”  of all driver proteins in a tumour extracted by your framework? This would be a very useful investigation in drug sensitivity prediction, and even better, in the tumour-immune-drug interactions modelling (see for instance https://doi.org/10.1101/2020.10.08.331678 and https://www.preprints.org/manuscript/202209.0012/v1)

Answer: We greatly appreciate your insight. These works will be thoroughly examined and incorporated into our future research.